# Co-Occurrence of Wing Deformity and Impaired Mobility of Alates with Deformed Wing Virus in *Solenopsis invicta* Buren (Hymenoptera: Formicidae)

**DOI:** 10.3390/insects14100788

**Published:** 2023-09-27

**Authors:** Godfrey P. Miles, Xiaofen F. Liu, Esmaeil Amiri, Michael J. Grodowitz, Margaret L. Allen, Jian Chen

**Affiliations:** 1Biological Control of Pests Research Unit, United States Department of Agriculture-Agricultural Research Service, 59 Lee Road, Stoneville, MS 38776, USA; godfrey.miles@usda.gov (G.P.M.);; 2Delta Research and Extension Center, Mississippi State University, 82 Stoneville Road, Stoneville, MS 38776, USA

**Keywords:** *Solenopsis invicta*, deformed wing syndrome, bee deformed wing virus, pollinators, viral replication

## Abstract

**Simple Summary:**

Deformed wing virus (DWV) is a major honey bee pathogen found throughout the world. DWV, in association with the varroa mite, causes wing deformity, a shortened abdomen, and neurological impairments, leading to the mortality of millions of honey bee colonies worldwide. At least 12 ant species have been shown to harbor DWV, including the red imported fire ant, one of the most invasive and detrimental pests in the world. To date, there have been no reports in the literature of DWV causing symptoms in ants. In this study, we observed the classic honey-bee-like symptoms of deformed wings in laboratory and field colonies of the red imported fire ants and verified the presence and replication of DWV. This is the first report of the co-occurrence of DWV-like symptoms and DWV in ants. However, more research is needed to determine whether DWV is indeed the causative agent of DW syndrome in *S. invicta*.

**Abstract:**

Deformed wing virus (DWV), a major honey bee pathogen, is a generalist insect virus detected in diverse insect phyla, including numerous ant genera. Its clinical symptoms have only been reported in honey bees, bumble bees, and wasps. DWV is a quasispecies virus with three main variants, which, in association with the ectoparasitic mite, *Varroa destructor*, causes wing deformity, shortened abdomens, neurological impairments, and colony mortality in honey bees. The red imported fire ant, *Solenopsis invicta* Buren, is one of the most-invasive and detrimental pests in the world. In this study, we report the co-occurrence of DWV-like symptoms in *S. invicta* and DWV for the first time and provide molecular evidence of viral replication *in S. invicta*. Some alates in 17 of 23 (74%) lab colonies and 9 of 14 (64%) field colonies displayed deformed wings (DWs), ranging from a single crumpled wing tip to twisted, shriveled wings. Numerous symptomatic alates also exhibited altered locomotion ranging from an altered gait to the inability to walk. Deformed wings may prevent *S. invicta* alates from reproducing since mating only occurs during a nuptial flight. The results from conventional RT-PCR and Sanger sequencing confirmed the presence of DWV-A, and viral replication of DWV was confirmed using a modified strand-specific RT-PCR. Our results suggest that *S. invicta* can potentially be an alternative and reservoir host for DWV. However, further research is needed to determine whether DWV is the infectious agent that causes the DW syndrome in *S. invicta*.

## 1. Introduction

The red imported fire ant, *Solenopisis invicta* Buren (Hymenoptera: Formicidae), is among the 100 world’s worst-invasive alien species [1]. Native to South America, they have invaded many countries and regions [2] and become significant pests in the infested areas, due to their adverse impacts on human health, agriculture, wildlife, pets, and livestock [3,4]. The red imported fire ant can be a problem for honey bees and beekeepers. For example, in Texas, fire ants were often observed preying on bee brood and dead adult honey bees, particularly when the bee colonies were weak [5]. It is a great challenge to control fire ants, and their management heavily relies on synthetic insecticides. Due to the ever-increasing public concern about the potential adverse effect of synthetic insecticides, tremendous effort has been made in searching for safer alternatives. The utilization of their natural pathogens, such as viruses, as potential biological control agents, has been an active research area [6].

Deformed wing virus (DWV) is one of the most-intensively studied insect pathogens in the world due to its significance for the health of honey bees and other pollinators [7]. DWV negatively impacts honey bees, resulting in physical abnormalities, including wing deformities, shortened abdomens, discoloration of adult bees, and neurological impairments [7]. Its presence correlates with colony failure, particularly in association with the ectoparasitic mite, *Varroa destructor* [7,8,9]. DWV is a positive-sense, single-stranded RNA virus belonging to the genus *Iflavirus* in the order *Picornavirales* [10,11] and is a quasispecies virus with three main variants (A, B, and C), found in many parts of the world, including at least 32 U.S. states [12]. To date, DWV has been detected in 65 arthropod species in eight insect orders and three Arachnida orders [7]. DWV is known only to induce wing deformities in honey bees, bumble bees, and wasps [7,13,14]. At least 12 ant species have been shown to harbor DWV, including *Solenopsis invicta* (red imported fire ant) [11,12] with replication found in only a few ant species, including *Linepithema humile* [10,15] and *Myrmica rubra* [16]. There have been no reports of a replicative form of DWV causing any visible pathogenic symptoms in ants as previously described in honey bees, bumble bees, and wasps (e.g., deformed wings, ataxia, leg paralysis, or body discoloration).

This study had three main objectives. Firstly, it aimed to document the observed classic honey-bee-like symptoms of deformed wings in *S. invicta*. This was achieved through a detailed description, supplemented with still images and videos of DW alates. The second objective was to demonstrate the co-occurrence of DWV with DW alates of *S. invicta* by analyzing the DWV in not only alates with deformed wings, but also workers from the same colonies. The final objective was to verify the presence of the replicating form of DWV in *S. invicta* workers and both asymptomatic and symptomatic alates.

## 2. Materials and Methods

### 2.1. Ant Colony Collection and Maintenance

A total of 43 *S. invicta* colonies were used for this project. Twenty-seven colonies were collected and maintained in the laboratory, and 14 colonies were used in situ—left in the ground and sampled for various assays. Two colonies were initiated using new queens collected on 21 July 2021, in the parking lot of Nelco Cineplex, Greenville, Mississippi (see Appendix A in the Appendix A for the information on each colony). For the laboratory colonies, ants were separated from mound soil using a modified dripping method [17], unless otherwise stated. All but two lab colonies were maintained at 26 ± 2 °C and 50% RH in Fluon-coated (Insect-a-Slip, Rancho Dominguez, CA, USA) plastic trays (55 cm × 44 cm × 12 cm) and given ad libitum access to food and water, which consisted of frozen crickets, 10% sugar water solution, and distilled water. Two lab colonies were provided with a finely ground ant food consisting of dried banana, granola bar, and frozen crickets in a 0.5:0.5:1.0 ratio.

### 2.2. RNA Extraction, cDNA Synthesis, and Reverse-Transcriptase PCR

Workers (20 to 50 mg fresh weight; 10 to 20 ants), individual alates (single, 5 to 11 mg fresh weight), or pooled alates (30 to 50 mg fresh weight; 5 alates) were added into microcentrifuge tubes separately and stored at −80 °C until RNA extraction. Micro-pestles (Eppendorf, Enfield, CT, USA) were used to homogenize the samples, and total RNA was extracted from each sample using the Zymo Direct-zol™ RNA MiniPrep Kit (Zymo Research, Irvine, CA, USA) following the manufacturer’s guidelines. The extracted RNA was treated with DNAse I to remove contaminating genomic DNA following the manufacturer’s directions. One microgram of total RNA from each sample was used for cDNA synthesis by the SuperScript™ IV First-Strand Synthesis System (Invitrogen Inc., Carlsbad, CA, USA) following the manufacturer’s instruction. RT-PCR was performed with sets of primers, found in Table 1, to detect DWV and the genetic variant, DWV-A.

Primers used for this study: At first, we used the DWV, DWV-A, and DWV-B sets of primers for random hexamers’ cDNA. We detected the presence of DWV-B in a single sample, so we did not keep using this set of primers; the DWV primer set generates an amplicon of 139 bases, which is difficulty to obtain good quality Sanger sequences from; instead, we switched the DWV-6F and B8 primer set, which generates an amplicon of 393 bases. Later, when we detected the replication of DWV, we used the DWV-F15 and B23 primer set based on the reference. This is a gene-specific primer set and generates an amplicon of 451 bases. We used the tagged DWV-F15 strand (gene-specific) primer for RT-cDNA and PCR with the tag and DWV-23B to determine the replication of DWV. We used the DWV-15F and B23 primer set to detect DWV from *S. invicta*. See Table 1 for the primer sequences and references.

### 2.3. Sanger Sequencing

The amplicons of 5 µL RT-PCR were electrophoresed in 2% agarose gel containing 0.5 µg/mL ethidium bromide and visualized under UV light. Then, the single-band (targeted) amplicons were purified using the ExoSAP-IT kit (Applied Biosystems™, Waltham, MA, USA). For the Sanger sequencing [18], we used the BigDye^®^ Terminator v3.1 Cycle Sequencing Kit (Applied Biosystems, Foster City, CA, USA). Sequencing reactions were performed in a 96-well plate cycling at 95 °C for 3 min, followed by 25 cycles of 95 °C for 30 s, 55 °C for 1 min, and finally, 68 °C for 2 min on a C 1000 Touch ^TM^ Thermal Cycler (Bio-Rad, Hercules, CA, USA). Post-sequencing reaction products were purified by ethanol/EDTA precipitation and injected into a 3730 XL DNA analyzer with Data collection 5.0 vision, Dye set Z (Applied Biosystems, Foster City, CA, USA). The positive control (extracted DWV from symptomatic worker honey bees) and negative control (NTC) were performed in each plate run. The sequencing results were analyzed using DNASTAR SeqMan Ultra vision 17.1 (Madison, WI, USA), and consensus and contig sequences were compared against known DWV viral sequences in GenBank by BLAST [19] to confirm viral identity.

### 2.4. Detection of DWV Replication in Solenopsis invicta

To determine the active replication of DWV in *S. invicta*, a modified two-step, strand-specific RT-PCR was performed [5]. Five hundred nanograms of total RNA from each sample was used to synthesize cDNA using Maxima Reverse Transcriptase (Thermo Fisher Scientific, Denver, CO, USA). This thermostable reverse transcriptase was used to minimize nonspecific priming. We employed a gene-specific primer coupled to a 5′ non-viral tag [20]; see Table 1. In brief, the primer binding took place at 65 °C for 5 min, and then, the reverse transcriptase reaction temperature was 65 °C for 30 min to reduce the secondary structure and to improve specificity. Negative controls including no template (water), no primer, no transcriptase, and the positive control (DWV symptomatic honey worker bees) were included in each set of RT-PCR reactions. The subsequent PCR amplification was carried out using a primer pair consisting of the tag only together with a virus-specific upstream primer, DWV-R B23; see Table 1, reviewed in [21]. This RT-PCR amplicon refers to the viral-RNA-dependent RNA polymerase. Briefly, 5 µL of cDNA from each sample was treated with ExoSAP-IT (Applied Biosystems, Waltham, MA, USA) to clean up the cDNA. PCR reactions were performed with Phusion High-Fidelity DNA Polymerase (New England BioLabs, Ipswich, MA, USA). PCR was briefly conducted at 98 °C for 30 s, followed by 35 cycles at 98 °C for 10 s, 55 °C for 30 s, and 72 °C for 30 s, with a final 72 °C extension for 10 min. The PCR products were visualized in 1.5% agarose gel containing 0.5 µg/mL of ethidium bromide. PCR products with a 451 bp size were purified with ExoSAP-IT (single band at 451 bp) or excised (with unspecific bands) and, subsequently, gel-purified with the QIAEX II Gel Extraction Kit (Qiagen, Germantown, MD, USA/Hilden, Germany). We also used a ten-times dilution of the template to minimize the chance of non-strand-specific cDNA via participation of the residual tagged cDNA primer [22]. The Sanger sequencing process was the same as previously described.

### 2.5. Videos of DW and NW S. invicta Alates

Eight videos depicting both DW and NW male and female alates were captured from six lab colonies and two field colonies. All videos were captured using a Keyence VHX 5000 (Itasca, IL, USA), except for one, where a Samsung Galaxy J7 Star mobile phone was used.

## 3. Results and Discussion

### 3.1. Symptom of DW Alates of S. invicta

From November 2021 to January 2023, 29 colonies were collected in Washington, Co, MS; 23 colonies produced alates, of which 17 colonies (74%) produced alates with visible wing deformities. We sampled an additional 14 colonies (NBCL 1-to-14) with alates (larvae, pupae, and adults) from August 2022 to June 2023, 9 of which had DW alates present (65%) (see Appendix A in the Appendix A for colony details). Wing deformity severity ranged from a single crumpled wing tip to fully crumpled wings for both male and female alates (Figure 1 and Figure 2, respectively). Figure 1B represents the degree of wing deformity most often found in the population of DW male alates thus far observed, with a portion showing more severe wing deformity, as in Figure 2C, or only wing tip deformity, as seen in Figure 1C. Additionally, Figure 1D shows a melanized male alate displaying deformed wings (top) compared to a specimen with normal wings, both from the same colony collected at the same time. We observed DW as early as the non-melanized pupal stage in male alates (images not shown).

The number of alates displaying DW in individual laboratory colonies varied greatly, ranging from 1.4% to 15%. The percent DW for both male and female alates is based on a specific period and does not represent percentages for the lifetime of the lab colony. Daily observations of adult alates were made throughout the summer and autumn of 2022 and April to June of 2023. The percent DW data can be found in Table 2. Laboratory Colony 8 had the greatest percent change in DW alates: From 21 August 2022 to 27 September 2022, 606 female alates were collected with 6 showing wing deformity (~1%) and 11 male alates with 1 with DW (9.1%). Then, interestingly, from 15 October 2022 to 29 November 2022, 118 male alates were collected, 39 with a wing deformity (33%), and no DW female alates were observed from Colony 8 during the latter time.

The above-mentioned Colony 8 and other colonies including Colony 2021, Colony 10, Colony 12, Colony 14, and Colony 1B (see Appendix A in the Appendix A for colony details), which were found to have the highest proportion of severely deformed wing male alates, died out before colonies where only a slight wing deformity was identified. For Colony 8, within a month after recording a high number of DW male alates, the colony died out except for a few hundred adult workers. In [15], it was found that DWV and *Linepithema humile* virus 1 were both replicating in *L. humile*, which suggests that they are not merely being vectored as viral particles, but potentially infecting their ant hosts, which makes these viruses candidates for the observed population declines seen in *L. humile* [23,24].

In addition to wing deformity, we identified many male DW alates displaying impaired mobility ranging from a slow, wobbly (ataxic) gait to the inability to walk or stand. Impaired mobility was observed in alates with varying levels of wing deformity. We identified six male alates with severe wing deformity (crumpled, stubby wings, <1 mm in length); however, they displayed no impaired mobility. On the other hand, we observed several male alates with altered mobility presenting with deformity of a single wingtip. Interestingly, many DW male alates presented a noticeably slower gait, but not all, compared to healthy NW alates reared in the same colonies as DW alates, as determined by multiple observers. The slow, wobbly (ataxic gait) is plainly visible in the Appendix A. Although the specimen number was small, 38 specimens, female alates did not display noticeably impaired mobility. In addition to female alates not showing noticeably impaired mobility, no workers were observed to have an altered mobility or leg paralysis. In Colony 8 (see above), 14 of the 39 DW male alates had some level of altered mobility (legs) and a single alate that was unable to move its wings. We identified four male alates who were able to move all legs, but unable to stand.

Eight videos show both male and female alates displaying both wing and leg deformities concomitant with altered mobility. All videos are present in the Appendix A Section, along with a detailed description of the contents of each video.

### 3.2. Identification of DWV in S. invicta DW Alates and Workers

Molecular analysis was performed to determine the presence of DWV and distinguish its variants in the *S. invicta* sample using DWV-specific primers, general primers [25], and primers specific to DWV-A [26]. We also used primers specific to DWV-B and were able to detect this subvariant in a single DW male alate from Colony 14, whereas Subvariant A was detected in multiple colonies. Since Subvariant A gave more-consistent results (more easily detectable in *S. invicta* colonies) compared to that of Subvariant B, we decided, for the purposes of this study, to use Subvariant A as the focus for this paper. Ants from 13 colonies were analyzed. The results are summarized in Table 3. DWV can occur in alates of both sexes, no matter if the alates exhibited wing deformity or not. Workers were DWV-positive in 10 of 13 colonies. Workers from two colonies started from new queens (G-1 and G-2), and the queen in Colony G-1 was also positive. We did not determine if DWV was present in every colony, nor did we look at the same castes from each colony. All Sanger sequencing results for DWV in *S. invicta* are contained in Appendix A.

Wing deformity and crippling paralysis are linked to deformed wing virus (DWV) in numerous pollinator species [7,9,14]. DWV has been identified in at least 12 ant species [10,27], including *S. invicta* [5]. However, none of the typical deformities associated with DWV in honey bees have been documented in DWV-positive ant species. This is the first report of the co-occurrence of DWV-like symptoms in ants and DWV.

### 3.3. Identification of the Replication Form of DWV in S. invicta DW Alates and Workers

To determine the presence of the replicative form of DWV in *S. invicta*, a modified two-step, strand-specific RT-PCR was performed [20,22] (see Section 2 and Appendix A, located in the Appendix A). Viral replication of DWV can be present in workers and male and female alates with and without the DW phenotype (Table 4). We did not examine other castes for virus replication. A representative DNA gel image (Figure 3) indicated the detection of the replicative form of DWV. The Sanger sequencing results were then compared to known DWV viral sequences in GenBank by BLAST, which identified the presence of DWV in all samples sequenced; the replicative form of the virus was also detected in several samples, including DW male and female alates, NW male alates, DW non-melanized pupae, worker pupae, and adult workers (Appendix A). We prepared a phylogenetic tree (Appendix A) that was based on the sequencing data of the RNA-dependent RNA polymerase gene from the DWV replication data (16 February 2023; see Sanger sequencing data) of *S. invicta*.

The replicative form of DWV was also detected in NW alates in addition to DW alates. The finding that NW alates also carry the replicative form of DWV mirrors what other research groups have found in honey bees [28,29], bumble bees, *Bombus terrestris*, [30], and the wasp, *Vespa crobro* [31]. For the first time, the presence of replicative DWV in *S. invicta* has been confirmed.

Although the specimen number was small, 38 specimens, we were unable to detect DWV replication in NW female alates. This finding, in addition to the absence of ataxia and leg paralysis in DW female alates, associated with a lower percent of female alates displaying deformed wings, suggests a possible sex-linked association. Additional work is needed to determine if this is the case.

Replication has been identified in only a few ant species, including *Linepithema humile* [10,15] and *Myrmica rubra* [16]. The only negative impact on the overall fitness of any ant species harboring DWV is differential gene expression of several immune response genes in *L. humile* [32]. To date, there have been no reports of the replication of DWV causing any outward morphological and behavioral changes in ants, as previously described in other pollinators (e.g., deformed wings, ataxia, leg paralysis, or body discoloration).

We are not aware of any sex-link between this virus and any other insect including ants. With regard to honey bees, a recent study by [33] (and the references therein) reported that emerging drones exhibited overt developmental deformities similar to those seen in the worker brood when injected with DWV as white-eyed pupae. They also reported that a percentage of drones without any outward deformities carried very high titer levels equivalent to drones with outward deformities.

## 4. Conclusions

We observed the classic DWV-induced symptoms found in DWV-infected honey bees, specifically deformed wings and impaired mobility, in laboratory and field colonies of *S. invicta* and subsequently verified the presence of DWV (and the replicative form of DWV) in the symptomatic and asymptomatic individuals using both RT-PCR and Sanger sequencing. This is the first report of DWV-like symptoms in ants and with the co-occurrence of replicating DWV. More research is required to gain an understanding of the DW phenotype observed in the DWV-positive alates of *S. invicta*, namely a direct causal link between DWV and observed wing deformity in *S. invicta*. DW alates are unable to fly, which is a barrier to nuptial flight and necessary in *S. invicta* reproduction; therefore, wing deformity can potentially impact populations of *S. invica*.

## Figures and Tables

**Figure 1 insects-14-00788-f001:**
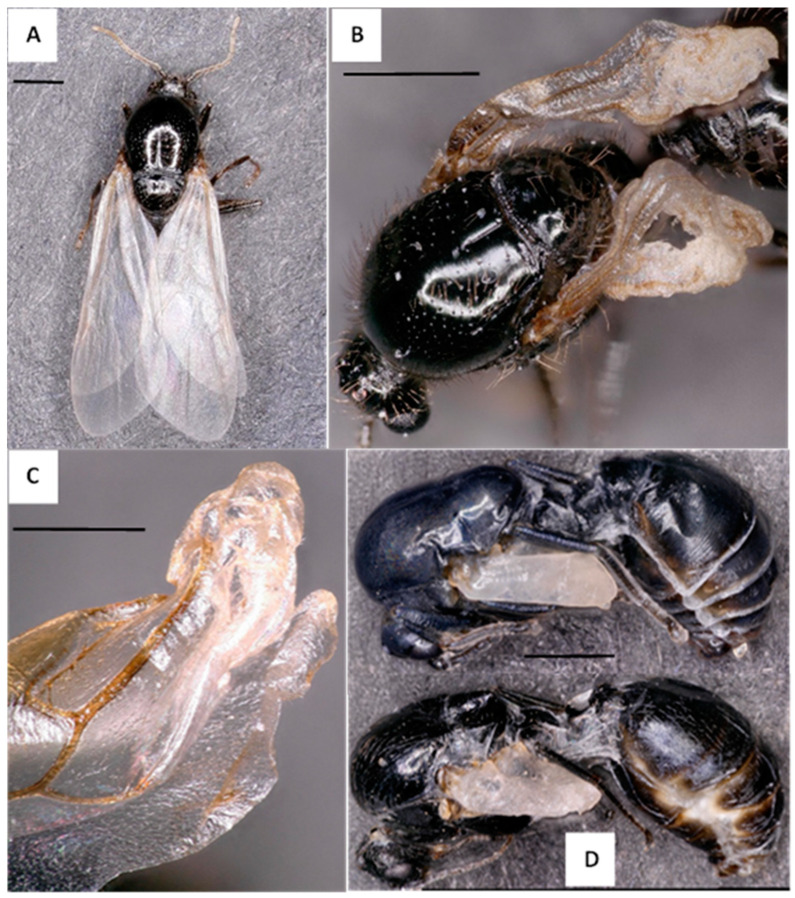
*Solenopsis invicta* specimens displaying the normal wing (NW) and deformed wing (DW) phenotype. (**A**) Normal wing (NW) adult male alate collected from Colony 8. (**B**) DW male alate from Colony 2021, which was the first DW alate identified. (**C**) Wing tip deformity of a DW male alate collected from Colony 11. (**D**) Two melanized male pupae with DW (bottom) and NW (top), both collected from Colony NBCL-1 collected directly from the ground. All images were captured using a Keyence VHX 5000 (Itasca, IL, USA). Scale bars are 1 mm for all images except Image C, which represents 0.5 mm.

**Figure 2 insects-14-00788-f002:**
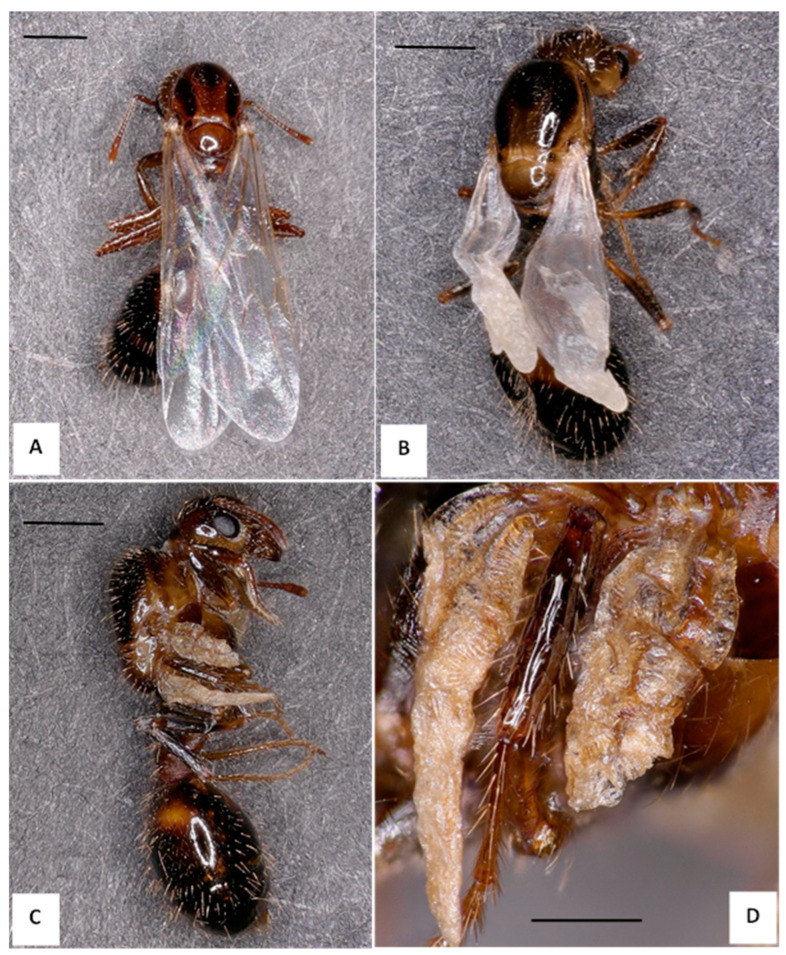
*Solenopsis invicta* specimens displaying the normal wing (NW) and deformed wing (DW) phenotype. (**A**) Normal wing (NW) adult female alate collected from Colony 8. (**B**) DW female alate with a moderate level of wing deformity, also collected from Colony 8. (**C**) Female DW alate with severe wing deformity, from Colony NBCL-14. (**D**) The same female alate as C under higher magnification. All images were captured using a Keyence VHX 5000 (Itasca, IL, USA). Scale bars are set at 1 mm for all images, except Image D, which represents 0.5 mm.

**Figure 3 insects-14-00788-f003:**
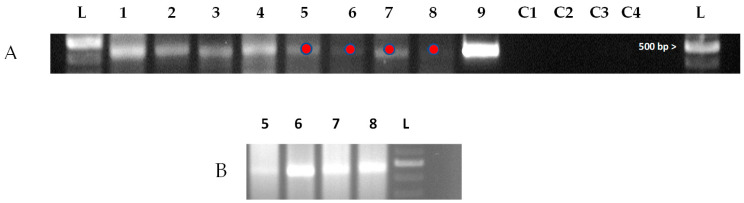
Replicative form of DWV. (**A**) Representative DNA agarose gel image showing detection of replicative form of DWV using primers specific to the RNA-dependent RNA polymerase (RdRp) gene of DWV in *Solenopsis invicta*. Lane order is as follow: (L) 100 bp ladder, (Lane 1) adult workers, (Lane 2) worker pupae, (Lane 3) deformed wing (DW) male alates (pooled, *n* = 5), and (Lane 4) male DW alate (different colony than Lane 3). Lanes 5–8 represent a 1/10 dilution of Lanes 1–4, respectively, (9) DWV-symptomatic Western honey bee (*Apis mellifera*), (C1) NTC, (C2) no primer, (C3) no transcriptase, and (C4) PCR-NTC. Red dots represent the location where a portion of the band (plug) was removed for reamplification. Amplicon length is 451 bp, 1.5% gel. (**B**) Reamplification of 1/10 dilution samples. Sanger sequencing results of the 1/10 diluted samples were all positive for the DWV RdRp gene; see Appendix A.

**Table 1 insects-14-00788-t001:** Primers used to detect DWV, DWV-A, and DWV-B.

Target	Primer Name	Primer Sequence	Product Size (bp)	Ref.
DWV	Forward	5’-GAGATTGAAGCGCATGAACA-3’	130	25
	Reverse	5’-TGAATTCAGTGTCGCCCATA-3’		
DWV-A	Forward	5’-GTCTTGTGGATGAAGGTTATATAACTGG-3’	168	26
	Reverse	5’-TCCGTAGAAAGCCGAGTTG-3’		
DWV	DWV-F F6	5’-TTTCCAGGTCCATTCCCCTATC-3’	393	14
	DWV-R B8	5’-TCATTCGCCTTACGACGGTTAG-3’		
DWV	DWV-F F15	5’-agcctgcgcaccgtggTCCATCAGGTTCTCCAATAACGG-3’	451	9
	DWV-R B23	5’-CCACCCAAATGCTAACTCTAACGC-3’		
	Tag	5’-agcctgcgcaccgtgg-3’		
DWV-B	DVQ_DWV-B-F	5’-ACCAACGCGTGTCGTTCCTG-3’	108	26
	DVQ_DWV-B-R	5’-ACAAGTGGTTGGTCCCGTCG-3’		

**Table 2 insects-14-00788-t002:** Percentage of DW male and female alates in *S. invicta* laboratory colonies collected over a specific period during 2022–2023 *.

Colony	Collection Period	Male Alates	DW, (%)	Female Alates	DW, (%)
2021	28/07/2022–12/09/2022	89	9, (10)	0	0
1	12/08/2022–26/08/2022	122	7, (5.7)	0	0
5	24/08/2022–26/09/2022	42	6, (14)	398	0
7	21/08/2022–27/09/2022	657	9, (1.4)	32	0
8	21/08/2022–27/09/2022	11	1, (9.1)	606	6, (1)
8 *	15/10/2022–29/11/2022	118	39, (33) *	0	0
9	17/08/2022–13/09/2022	55	4, (7.5)	0	0
10	22/08/2022–26/08/2022	14	2, (14.3)	425	0
12	9/08/2022–6/09/2022	73	11, (15)	987	0
14	9/08/2022–30/08/2022	139	15, (10)	4	0
1B	2/09/2022–27/09/2022	325	14, (4.3)	22	4, (18)
3B	4/09/2022–26/09/2022	33	4, (12)	385	0
5B	4/09/2022–14/10/2022	36	4, (11)	0	0
1C	12/04/2023–23/06/2023	83	11, (13.25)	0	0
3C	12/04/2023–22/06/2023	113	6, (5.3)	9	0
5C	12/04/2023–27/06/2023	46	3, (6.5)	11	0

* All lab colonies were collected in the vicinity of (33.160 N, 90.920 W). All alates from each colony were removed and counted on multiple collection dates within the collection period.

**Table 3 insects-14-00788-t003:** DWV detection in various castes from colonies of *S. invicta*.

Colony	Sample Type	Replicates	DWV Detected?	Colony	Sample Type	Replicates	DWV Detected?
1B	DW male alate	1	Yes	9	DW male alate	1	Yes
		2	Yes			2	Yes
	NW male alate	1	Yes		Worker	1	No
		2	Yes			2	Yes
	Worker	1	Yes	10	DW male alate	1	No
		2	Yes			2	Yes
		3	No		Worker	1	Yes
NCBL	DW male alate	1	No			2	No
		2	Yes			3	Yes
		3	Yes	11	DW male alate	1	No
	DW female alate	1	No			2	No
		2	Yes			3	No
	NW male alate	1	Yes		Worker	1	Yes
		2	Yes			2	No
	NW female alate	1	Yes	12	DW male alate	1	No
	Worker	1	Yes			2	No
		2	No			3	No
		3	Yes		Worker	1	No
		4	Yes			2	No
2021	Worker	1	Yes	14	DW male alate	1	Yes
		2	Yes			2	Yes
		3	Yes		NW male alate	1	Yes
4	DW male alate	1	No			2	Yes
		2	No		Worker	1	No
		3	No			2	Yes
	DW female alate	1	No			3	Yes
		2	No	G1	Worker	1	Yes
		3	Yes			2	Yes
	Worker	1	No	G2	Queen	1	Yes
		2	No		Worker	1	Yes
8	DW male alate	1	No			2	No
		2	Yes				
		3	Yes				
	DW female alate	1	Yes				
		2	No				
		3	Yes				
	NW male alate	1	No				
		2	Yes				
		3	Yes				
	NW female alate	1	No				
		2	Yes				
	Worker	1	Yes				
		2	No				
		3	Yes				
		4	Yes				

**Table 4 insects-14-00788-t004:** Detection of DWV replication in various specimens from colonies of *S. invicta*.

Colony	Caste	Replication Dilution	Is Replication Detected?
1B	Worker	2X dilution	Yes
1B	NW male alate	2X dilution	Yes
NBCL	Worker	2X dilution	Yes
2021	Worker	Not diluted	No
Colony 4	Deformed wing female alate	2X dilution	Yes
Colony 8	Deformed wing male alate	10X dilution	Yes
Colony 8	Deformed wing female alate	2X dilution	Yes
Colony 8	Normal wing male alate	10X dilution	Yes
Colony 8	Pupa of male alate	10X dilution	Yes
Colony 8	Worker	10X dilution	Yes
Colony 9	Worker	Not diluted	No
Colony 10	Worker	2X dilution	Yes
Colony 10	Pupa of Workers	10X dilution	Yes
Colony 11	Worker	Not diluted	No
Colony 12	Worker	Not diluted	No
Colony 14	Normal wing male alate	2X dilution	Yes
Colony 14	Worker	Not diluted	No
Colony G1	Worker	Not diluted	No
Colony G2	Worker	Not diluted	No

## Data Availability

All study data are included in the article and/or the SI Appendix. The data presented in this study are available upon request from the corresponding author.

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
