# Peer review of "Co-Occurrence of Wing Deformity and Impaired Mobility of Alates with Deformed Wing Virus in Solenopsis invicta Buren (Hymenoptera: Formicidae)"

_insects, 2023, doi:10.3390/insects14100788_

Round 1

Reviewer 1 Report (Previous Reviewer 1)

The authors have revised the manuscript, added more details on the study design, and clarified their objectives. They also added results regarding the prevalence of DW alates in each nest. Yet, the data presentation still needs improvement, and although the authors' replies to my comments include valuable information, these were not always added to the main text.

As impaired mobility is in the title and a main result, the analysis of the mobility data should be explained.

Information given in the answer to question #7 or #9 from the previous round should be included in the main text. For example, only DW alates were used in molecular testing. Same with #10; the answer states that primer information was added, but it’s not in the main text.

Figure 3: C4 not marked in the image.

Table 3 & 4 captions are very short and don’t include explanatory information.

L133 missing a word?

L162: Remove ‘from’

Author Response

Reviewer #1

Comments and Suggestions for Authors

The authors have revised the manuscript, added more details on the study design, and clarified their objectives. They also added results regarding the prevalence of DW alates in each nest. Yet, the data presentation still needs improvement, and although the authors' replies to my comments include valuable information, these were not always added to the main text.

As impaired mobility is in the title and a main result, the analysis of the mobility data should be explained.

Answer: The differences in mobility of DW alates compared to NW alates was performed by multiple observers. We did not use a grid walking test to evaluate the altered mobility observed in DW alates. This is something we can use in future work. Additionally, one can observed mobility differences between DW and NW alates from the multiple videos.

Information given in the answer to question #7 or #9 from the previous round should be included in the main text. For example, only DW alates were used in molecular testing. Same with #10; the answer states that primer information was added, but it’s not in the main text.

Answer: We have added additional text to the appropriate locations throughout the manuscript.

Primer text: At first, we used DWV, DWV-A, and DWV-B sets of primers for random hexamers cDNA. We detected the presence of DWV-B in a single sample, so we did not keep using this set of primers; DWV primer set generates an amplicon of 139 bases, which is difficult to get good quality Sanger sequences; instead, we switched DWV-6F and B8 primer set which generates an amplicon of 393 bases. Later, when we detect replication of DWV we used DWV -F15 and B23 primer set based on the reference. It is gene-specific primer set and generates an amplicon of 451 bases. We used the tagged DWV-F15 strand (gene-specific) primer for RT-cDNA, and PCR with tag and DWV-23B to determine replication of DWV. We used DWV-15F and B23 primer set for detect DWV from S. invicta. See Table 1 for primer sequences and references.

Figure 3: C4 not marked in the image.

Answer: The lane identifier “C4” has been placed back with the figure.

Table 3 & 4 captions are very short and don’t include explanatory information.

Answer: We have added text to both table captions to better explain the table contents. Please see table captions.

 L133 missing a word?

Answer: Corrected.

 L162: Remove ‘from’

Answer: Corrected.

Reviewer 2 Report (New Reviewer)

I am providing evaluation on the revised version of the article titled “Co-occurrence of Wing Deformity and Impaired Mobility of Alates with Deformed Wing Virus (DWV) in Solenopsis invicta Buren (Hymenoptera: Formicidae)”. After carefully reading the revised manuscript, I feel the authors have done a nice job addressing my concerns, especially that they have toned down DWV being the causative agent for wing deformity in the invasive fire ant.

I, however, have two additional suggestions:

1. I have a difficult time locating the Discussion section (3. Results, and then 4. Conclusion). Shouldn't it be 3. Results and Discussion?

2. The authors may want to discuss a bit on why WD is preferentially taking place in male alates over female alates – also the authors could use a statistical analysis to test if the difference is significant.

Below are some more minor suggestions:

Line 169: “but subvariant A gave more consistent results than subvariant-B” Please clarify what “more consistent results” is.

L244 but potentially “parasitizing” their ant hosts – replace it with “infecting”

L294-298 This paragraph looks to me more like something that should go to Discussion, may want to consider to remove it or move it to Discussion (but referring to major comment #1)

L307-308 There seems to be a few lines hidden behind Figure 3.

L354 I believe the authors meant Replication “of DWV”…

Throughout the manuscript, quite a few S. invicta need to be italicized.

Not sure where C4 is in Figure 3. Please indicate.

A few typos and format errors remain. I would suggest a detailed proofreading before finalizing the manuscript.

Author Response

Reviewer #2

I am providing evaluation on the revised version of the article titled “Co-occurrence of Wing Deformity and Impaired Mobility of Alates with Deformed Wing Virus (DWV) in Solenopsis invicta Buren (Hymenoptera: Formicidae)”. After carefully reading the revised manuscript, I feel the authors have done a nice job addressing my concerns, especially that they have toned down DWV being the causative agent for wing deformity in the invasive fire ant.

I, however, have two additional suggestions:

  1. I have a difficult time locating the Discussion section (3. Results, and then 4. Conclusion). Shouldn't it be 3. Results and Discussion?

Answer: As with a few of the images and text, somehow the words “…and Discussion” were lost from the text. We have now fixed this issue.

  1. The authors may want to discuss a bit on why WD is preferentially taking place in male alates over female alates – also the authors could use a statistical analysis to test if the difference is significant.

Answer: We are not aware of any sex-link between this virus and any other insect including ants. With regard to honey bees, a recent study by Woodford et al. (2023), reported that emerging drones exhibited overt developmental deformities similar to that seen in worker brood when injected with DWV as white eyed pupae. They also reported a percentage of drones without any outward deformities carried very high titer levels equivalent to drones with outward deformities.  In ants, just like honeybees, females are diploid and produced from fertilized eggs, whereas males are haploid produced from unfertilized eggs. How this sex system is related to what we have observed will be a nice topic for future research.

Due to the extremely low occurrence of deformed wing syndrome in female alates, no statistical analysis is really needed.

Below are some more minor suggestions:

Line 169: “but subvariant A gave more consistent results than subvariant-B” Please clarify what “more consistent results” is.

Answer: (Line 269) We also used primers specific for DWV-B and were able to detect this subvariant in a single DW male alate from colony 14, whereas subvariant-A was detected in multiple colonies. Since subvariant-A gave more consistent results (more easily detectable in S. invicta colonies) compared to that of subvariant-B, we decided, for the purposes of this study, to use subvariant-A as the focus for this paper.

L244 but potentially “parasitizing” their ant hosts – replace it with “infecting”

Answer: This has been corrected.

L294-298 This paragraph looks to me more like something that should go to Discussion, may want to consider to remove it or move it to Discussion (but referring to major comment #1)

Answer: As with a few of the images and text, somehow the words “…and Discussion” were lost from the text. We have now fixed this issue.

L307-308 There seems to be a few lines hidden behind Figure 3.

Answer: We apologies for this. We are not sure why the images moved from their original locations.

L354 I believe the authors meant Replication “of DWV”…

Answer: This has been corrected.

Throughout the manuscript, quite a few S. invicta need to be italicized.

Answer: This has been addressed.

Not sure where C4 is in Figure 3. Please indicate.

Answer: The lane identifier “C4” has been placed back with the figure.

Comments on the Quality of English Language

A few typos and format errors remain. I would suggest a detailed proofreading before finalizing the manuscript.

Answer: We have gone through the manuscript and have correct all typos and formatting errors.

This manuscript is a resubmission of an earlier submission. The following is a list of the peer review reports and author responses from that submission.

Round 1

Reviewer 1 Report

Miles et al describe the occurrence of symptomatic deformed wing virus, an emerging honey bee pathogen, in Solenopsis invicta. The study finds alates (winged ants) with wing deformities (DW) in multiple Solenopsis invicta colonies and the DW phenotype correlates with lower adult body weight. Additionally, the authors test alates and workers for deformed wing virus. The virus is associated with wing deformities in honey bees, with occasional observations of this phenotype in other winged hymenopterans. Workers, DW alates and heathy alates all test positive for DWV and the negative strand intermediate, which indicates active DWV replication. The results are novel and interesting, especially as S. invicta is a highly invasive species with the potential to vector bee pathogens.

However, I have some major concerns about the methodology and presentation of the results. I think addressing these concerns would strengthen the link between the DW phenotype and DWV infections.

It the current form, the manuscript reads as if the authors have used opportunistic sampling and it is not clear which samples have been used in which part of the analysis, especially molecular analyses. More information on experimental design or sampling scheme is needed here. It should also be clarified which primers were used in the prevalence testing and those results should be presented. As prevalence results are currently only described for one colony. Some statistical testing (maybe simple chi-square?) would be good to see if DW/ normal wings and infected/uninfected are linked. From the manuscript it reads like there would be a significant link but this is not shown.

I found it surprising that active DWV replication was shown in all samples presented, again, how many samples were tested and how many DW/ normal wings test positive for replication. I think a table (maybe in the supplement if that’s preferred) could clear things up.

Results are only briefly discussed. Why would reduce weight and DW phenotype correlate? Lower weight of alates with wing deformities, but not in pupae. Could alates be fed less when works see they have wing deformities and does the colony stops investing into DW alates? How do you explain positive DWV replication in NW alates?

Minor comments:

L64-65: No reports of replicative DWV causing symptoms, but reports of DWV replication in ants has been shown before, please rephrase

L89: Weight comparison protocol. This section needs more detail, what was compared? Was weight compares within colonies?

L105: How many workers, alates and alate pools were used. Did you collect from all 33 colonies and how many samples were taken per colony? Did all samples show DW or were NW samples used in PCR too?

L115: With random primers?

L116: Were samples always tested for all 5 targets in table 1?

L124: Please add info on which primers were used for sequencing.

L134: Consensus and contig sequences? Please clarify

L142: Should reference here be 23 instead of 24?

L143: Was primer binding really 65°C?

L147-149: This sentence is repeated in L152, maybe delete here, as the sentence in L150 is on cleaning cDNA, which happened before PCR.

L165: RIFA, acronym should be spelled out once

L163: This section reads more like captions from the supplement and not like a section in the main doc. Could you describe what videos were recorded and why certain samples were chosen for recording?

L213: I can’t see methods describing how mobility data was obtained. Was the slower gait in DW than NW alates measured or just anecdotal?

L225: If daily observations were made this should be stated in the methods.

L223: Only results from colony 8 are described here, what percentage of DW alates were found in other colonies?

L235-L239: Again, a lot of this would fit into the methods section

L259: Where are the data? What percentage of tested alates were DWV-positive? Did positive RT-PCR results correlate with DW phenotype?

L:261-262: This sentence seems misplaced as the sentence above only reports detection of DWV and not of its replicative form.

L265: Viral replication was found in DW and NW workers and alates. Was that surprising and could you discuss why those NW alates test positive for replication.

Figure 3: All tested samples and categories show positive for the DWV negative strand.. How can you ensure that amplification does not result from residual tagged 

Reviewer 2 Report

Miles et al analyzed both lab and field Solenopsis invicta colonies for wing deformity in alates (female and male), a typical symptom of deformed wing virus (DWV) in honey bees, and attempted to establish the observed symptom with the presence (and replication) of DWV. The authors found varying proportions of DW alates in the colonies and also that DW alates possessed less body weight than normal wing (NW) alates. While they found the replicative form of DWV in DW alates, NW alates also tested positive for DWV with the presence of replicative from.

Although the topic is interesting, I would not be able to recommend publication of this work as it suffers some flaws.

  1. How many colonies were subjected to DWV RT-PCR assay or detection of replicative form? Are all surveyed colonies infected with DWV or with replicative form? Are all DWV-infected colonies producing DW alates? Are the proportions of DW alates correlated with the infection status? Without the information, I found it extremely challenging to establish the link between the observed symptoms and DWV infection, which is the key message this article wanted to convey.
  2. Another key concern is that the authors failed to follow Koch's Postulates to verify that the DW symptom is actually caused by DWV. I would inoculate uninfected fire ant colonies with DWV and test if the symptom recapitulate. Without such data, the author’s conclusion is not supported.
  3. Data presentation needs significant improvement or clarification. For instance, the two ladders (L) in Fig 3 are not at the same level (or are they even the same ladder?); what are those red dots for (Lane 5-8); no information regarding to colony identity for the samples included in Fig. 3.